# Cell Signaling Pathway Reporters in Adult Hematopoietic Stem Cells

**DOI:** 10.3390/cells9102264

**Published:** 2020-10-09

**Authors:** Jolanda. J.D. de Roo, Frank. J.T. Staal

**Affiliations:** Department of Immunology, L3-Q, Leiden University Medical Center, 2333 ZA Leiden, The Netherlands; j.j.d.de_roo@lumc.nl

**Keywords:** murine reporter models, cell signaling pathways, fluorescent reporter proteins, hematopoietic stem cell biology, murine multi-reporter

## Abstract

Hematopoietic stem cells (HSCs) develop at several anatomical locations and are thought to undergo different niche regulatory cues originating from highly conserved cell signaling pathways, such as Wnt, Notch, TGF-β family, and Hedgehog signaling. Most insight into these pathways has been obtained by reporter models and loss- or gain of function experiments, yet results differ in many cases according to the approach. In this review, we discuss existing murine reporter models regarding these pathways, considering the genetic constructs and reporter proteins in the context of HSC studies; yet these models are relevant for all other stem cell systems. Lastly, we describe a multi-reporter model to properly study and understand the cross-pathway interaction and how reporter models are highly valuable tools to understand complex signaling dynamics in stem cells.

## 1. Introduction

It is well-established that a small number of signaling pathways control the development of all cell types in the mammalian body [1]. The dosage, combinations, and timing of such major signaling pathways determine the cell fate developmental decisions that are being made by complex cell and organ systems. During development and regeneration, the stem cells of a given organ or tissue type are crucially important for further growth and homeostasis. Hematopoietic stem cells (HSCs) are specialized cells with the capacity to maintain the homeostatic equilibrium by either self-renewal or differentiation into all blood-borne cell lineages. These two important characteristics are regulated by molecular mechanisms employed by complex crosstalk between the HSCs and a specialized niche or microenvironment. HSCs reside in bone marrow niche that is composed out of different cell types, such as mesenchymal stem cells (MSCs), Cxcl12-abundant reticular cells (CAR), perivascular cells, endothelial cells, and osteoblasts [2]. Specific signaling pathways mediate cell–cell interactions by either direct binding of ligands to receptors or secreted factors to receptors in close vicinity of the receiving cell. Highly conserved signaling pathways have been reported to play an important role in HSC self-renewal, specification, and differentiation. Synergistic or antagonistic signaling networks consisting of pathways such as Wnt, Notch, TGF-β/BMP, and Hedgehog have been studied widely but mostly separately from each other. Several knock-out, knock-in, or transgenic reporter models have been useful tools to decipher little bits of a larger puzzle of the functioning of these signaling pathways. However, the orchestration and fine-tuning of the total cell communication network still needs to be unraveled. In this review, we discuss existing murine reporters of these highly conserved signaling pathways and possible advancements to improve its use for complex pathway analysis in HSCs. Lastly, we propose a transgenic murine multi-reporter strategy for simultaneous pathway activity measurement adapted to flow cytometry techniques.

## 2. General Cell Signaling in HSCs

Positional structure is particularly important for HSC development, heretofore evident at the exclusive residence of the first definitive HSCs during embryogenesis in the ventral area of the aorta [3]. Further hematopoietic stem cell development undergoes a strictly regulated journey along different anatomical sites following ontogenesis, such as yolk sac, placenta, fetal liver, spleen, and finally into the bone marrow. The initial HSC pool needs to supply the entire organism with sufficient postnatal HSCs, resulting in largely cycling cells undergoing symmetric divisions while still maintaining self-renewal capacity and multipotency mainly in the fetal liver [4]. However, when HSCs arrive at the bone marrow they suddenly become quiescent [5], indicating that the microenvironment might play an important role in adult HSC maintenance. At present, the general belief is that a specialized niche in the bone marrow conserves HSC multipotency, while differentiation is thought to be directed by a slightly altered composed niche [2]. The molecular cross-talk between HSCs and their niches, also known as the stem-cell-niche synapse is thought to influence HSC decisions and act as principal driver for hematopoietic homeostasis.

Although most in vitro and in vivo gain-of-function studies validate a role of conserved signaling pathways in HSC regulation, many in vivo loss-of-function models of these signaling pathways suggest that none of these pathways are indispensable for adult HSC function. Nonetheless, the complexity of niche interactions with HSCs and the complexity of intracellular signaling pathway cross-talk make the interpretation and combination of these results difficult and prejudiced. Rather, a combination between suppression and activation of these pathways will ultimately solve the balance of quiescence and proliferation in HSCs. Careful experiments will be needed to elucidate such effects; reporter systems for these pathways serve as invaluable tools in such analyses.

## 3. Reporter Systems

Reporter systems are useful models to visualize the genetic regulation of a cellular response. Most signaling pathways initiate a cascade by several intracellular signaling molecules that end up placing transcriptional activator or repressor proteins to bind at a specific DNA sequence. This binding enhances the rate of transcriptional initiation in the promoter region of selective target genes, which in many cases is a complex convergence of multiple signals on the same promoter. The discovery of transcriptional activators and target genes have offered researchers means to genetically build-in reporter molecules to measure signaling pathway activity. Various reporter assays ranging from stable cell lines, transduction vectors, target gene-knock-ins, or multimerized transcription factor binding sites have been employed, but can give different outcomes when studying the same pathway.

The most widely used detection systems in living organisms are bioluminescent proteins genetically engineered into reporter systems. The most commonly used reporter proteins in signaling pathway assays are firefly and bacterial luciferase (*luc* and *lux* gene, respectively) and jellyfish Green Fluorescent Protein (GFP). Nowadays, more advanced fluorescent proteins offer easier detection methods, expanding the possibilities of understanding quantitative and qualitative cell signaling. Nevertheless, caution should be taken with interpretation of reporter models mostly due to their genetic context. Barolo, commented on several important factors to consider when interpreting the data of Wnt/β-catenin/TCF reporters, which are in fact applicable to every reporter model [6]. Reporters which aim to study the same signaling pathway but show discrepant results could be caused by: (1) a differing genetic context leading to differential signal sensitivity, (2) the use of enhanced pathway specific DNA binding sites to increase signal sensitivity, (3) differential signaling modes due to gene regulatory functions outside the pathway. For instance, the random integration of DNA binding sites dismisses the genetic context of the insertional location and thus can affect the reporter sensitivity. Transcriptional regulation on the other hand, is an important molecular switch to control signaling pathway (de)activation and is regulated by DNA flanking regions which are lacking when inserting artificial DNA binding sites and in turn affect reporter sensitivity. Collectively, these points of consideration are also indications on how to improve existing reporter models for better thought-out genetic strategies. In the following sections, we will explain the existing Wnt, Notch, TGF-β/BMP, and Hedgehog cell signaling murine reporter models in HSC biology and carefully consider how to design a multi-reporter transgenic model considering the genetic, molecular, and reporter protein context.

## 4. Wnt Signaling

Wnt signaling is a highly conserved pathway with a prominent role in embryogenesis and adult stem cells. It is also known to play a decisive role in a variety of malignant and nonmalignant hematopoietic diseases. Canonical and non-canonical Wnt signaling have distinct functions, but both have formerly been reported in HSC regulation. In this review we will comment on the canonical Wnt signaling pathway, for which a diverse set of in vivo reporter models have been created and which has been much studied in the context of HSC homeostasis. A noteworthy live-cell noncanonical reporter, Wnt5-GFP-KIF26B, could be an interesting future noncanonical in vivo reporter to look forward to [7]. However, this reporter has not been used in any studies concerning hematopoiesis and will not be further discussed here.

Secreted Wnt proteins activate the signaling cascade by binding to their corresponding Frizzled receptors and LRP co-receptors, leading to the nuclear translocation of cytoplasmic β-catenin (Figure 1). Without Wnt activation, β-catenin levels are kept low by proteasomal degradation via the so-called destruction complex that is composed of Axin1, Axin2, tumor suppressor gene product (APC), casein kinase1 (CK1α), and glycogen synthase kinase (GSK-3β). Upon Wnt ligand-receptor binding, Axin is sequestered to the intracellular part of the activated receptor, leading to the inhibition of destruction complex formation and subsequent β-catenin ubiquitination for proteasomal degradation. In the nucleus, β-catenin binds to TCF/Lef, transforming them into transcriptional activators by detachment of the co-repressor Groucho. Consequently, target genes are activated, such as *cyclinD1* and *axin2* (*Conductin*), among others. Axin2 is well known for also being a negative regulator of the Wnt signaling pathway and plays an important role in signaling regulation [8].

Several canonical Wnt signaling reporters have intended to make Wnt signaling visible and understandable (Table 1). Most of the Wnt reporters are derivatives from the TOP-FLASH construct that was designed for transient transfection experiments [9], which exploit a series of multimerized TCF/Lef DNA binding sites randomly integrated in the genome. These reporters have been mainly used to study the role of Wnt signaling in embryogenesis and organ development, yet little research has been done on the hematopoietic system with these models. Over time, a variety of minimal promoters and alternating numbers of binding sites have been tested with a LacZ reporter protein, resulting in newly found Wnt-active locations in every consecutive study. Each study claims to have a better reporting mouse model, yet comparison is hardly possible due to different genetic backgrounds, promoters, and research purposes [10,11,12]. Ferrer-Vaquer, opted for a distinct approach by changing *LacZ* for a *GFP* fluorescent protein in the already existing TCF/Lef-LacZ and BAT-gal reporters, but was only successful in the TCF/Lef-LacZ derived variant. They steadily proposed that their new TCF/Lef:H2B-GFP reporter expression is independent of the DNA integration site, even though no confirmative data contributed this claim [13]. Nonetheless, this reporter strategy does improve the ability to obtain quantifiable in vivo resolution and even cell tracking and cell division measurement due to the GFP stability offered by the H2B fusion protein. Interestingly, GFP expressing patches were detected in what are thought to be primitive erythroid cells at embryonic day 7.5 and GFP positive cells were visible in the postnatal thymus. A distinct approach derived from the TOPGAL reporter [10,14] are the ins-TOPEGFP and ins-TOPGAL reporter mice which were designed with β-chicken globin HS4 insulators to minimize positional genetic effects on the reporter expression [15]. These models showed to be useful for adult-tissue investigation and showed active Wnt signaling in mature splenic T cells. Nevertheless, the ins-TOP models are not available anymore even though they were the only Wnt reporters with the most commonly used C57BL/6 genetic background for HSC studies. A completely distinct genetic strategy was employed for the Ax2/d2EGFP reporter, which has a random insertion of the *Axin2* promoter, exon 1, and intron 1, reported by a reduced stable enhanced GFP [16]. Although the reporter activity seemed to be well detectable in developing organs, this reporter has been hardly used in other studies. In that same year, Lustig et al., constructed the first knock-in Conductin^+/lacZ^ reporter, to study colorectal and liver tumors [8]. As a result of the targeted integration strategy, the reporter model is additionally an *Axin2* heterozygote, but no negative effects were reported. The Conductin^+/lacZ^ reporter is considered to be the most reliable Wnt reporter to date [17], however the perceived disadvantage is its difficulty of use in live cells due to the administered substrate for LacZ detection, which requires harsh conditions such as hypo-osmatic loading. We recently developed an improved canonical Wnt signaling reporter with the same *Axin2* targeting construct but with an mTurquoise2 fluorescent marker: Axin2-mTurquoise2 [18]. This fluorescent protein facilitates single-cell resolution and quantitative reporter expression even in fragile cells such as HSCs and thymocytes. Our model has the advantage of avoiding lingering reporter protein expression as in the *TCF/Lef:H2B-GFP* reporter and respects the endogenous genetic context of the *Axin2* Wnt target gene. Lastly, the group of Roel Nusse, created Wnt reporters with cell lineage tracing capacity by adopting an *Axin2* responsive element with a Cre-recombining reporter cassette in the *Rosa26* locus [19]. The most recent Wnt reporter, the Axin2^P2A-rtTA3-T2A-3xNLS-SGFP2^ murine model, makes use of a knock-in multi-cystronic cassette upstream of the *Axin2* stop codon containing an rtTA3 doxycycline-inducible driver, a triple nuclear localization signal GFP fusion protein and self-cleaving 2A peptides, which preserves both functional alleles of the *Axin2* gene [20]. Notwithstanding the elegant design, this Axin2 reporter detection also suffers from lowly detectable reporter signaling as all fluorescent protein Axin2 reporter models and even hinders the full potential of their employed Wnt driven tetO-Cre strategy for lineage tracing. Overall, *Axin2* remains to be the best Wnt reporting gene, motivating for the generation of smarter transgenic models to overcome the low-level and low-amplitude oscillations of *Axin2* expression [21].

## 5. Notch Signaling

Notch signaling is a conserved pathway with functions in homeostatic and developmental processes. Both canonical and non-canonical Notch pathways have a relatively simple signal transduction without any signal amplification steps and are dependent on cell–cell interactions. Notch ligand-receptor binding activates proteolytic cleavage of the Notch receptor, releasing the Notch intracellular domain (NCID) (Figure 2). The NCID translocates to the nucleus, and binds together with DNA binding protein CBF-1 (also known as CLS or RBPJ) and co-activator Mastermind (MAML1) to initiate the transcription of target genes such as *Hes* and *Hey*. Nonetheless, Notch activity is highly context-dependent due to modulation through accessory proteins on the extracellular and intracellular side, resulting in elaborate receptor complexes. Various murine reporters have been published attempting to faithfully visualize Notch signaling (Table 1). The best-known models are the transgenic Notch reporter (TNR) [24] and the NAS (Notch Activity Sensor) reporter [25]. Both reporters make use of synthetic promoters and multiple *CBF* binding sites to improve reporter sensitivity. The TNR model is well known for the study of Notch signaling in HSC stem cell state [24]. By combining the Notch reporter with the Wnt reporter TOPGAL, a synergistic function for stemness maintenance was reported. This notion seems to be consistent with earlier studies claiming active Notch plays a role in enhanced self-renewal and decreased differentiation in hematopoietic progenitors [40,41]. Additionally, a direct correlation between active Wnt signaling and promotion of Notch signaling was reported through the expression of Notch target genes *Hes1* and *Dtx1* [24].

Less synthetic Notch endogenous murine reporters are available, where either Notch related promoters or directed knock-in strategies have been used respectively [27,28]. The *Hes1*-EmGFPSAT mice, which is a knock-in of the *Hes1* region, showed real-time Notch activity in the intestine through administration of a pharmacological Notch inhibitor dibenzazepine (DBZ) to show GFP abolishment in intestinal crypts, but residual signaling in the villi. Oh et al., used the same *Hes1*-EmGFPSAT model together with tamoxifen-inducible CreER knock-in mice for individual Notch receptors to study Notch signaling in hematopoiesis [42]. They showed a prominent role for Notch signaling through the Notch2 receptor and *Hes1* expression in long term HSCs. Additionally, they reported molecular priming towards the erythrocytic lineage in *Hes1* expressing Lin^neg^/cKit^+^/Sca1^+^ cells, whereas Notch negative cells tend to differentiate towards megakaryocytic and granulocyte-monocytic subsets. Alternatively, the *Hes1*-GFP reporter [27] was used to study Notch signaling in bone turnover and bone marrow, showing relatively little GFP expression in bone histological sections [43].

The most recent Notch reporter is a derivative from the TNR reporter with a human histone H2B–Venus (YFP) fusion fluorescent protein; CBF:H2B-Venus [26]. This model claims to be an improved tool for in vivo imaging, which might be true for Notch-regulated morphogenic embryogenesis studies. However, the authors self-criticize that a better fluorescent strategy could be chosen due to the delayed Notch signal of the histone-fluorescent protein fusion and perdurance of Venus. In this case, the Hes5-1 viral vector reporter used in chick neurogenesis [44] benefitted a better strategy with destabilized nuclear fast degrading Venus and a PEST sequence for fast degradation of the fluorescent protein. Even so, the CBF:H2B-venus model might still be interesting for short term Notch signaling tracking and cell division in HSC biology as well as lineage tracing studies from cells that have previously undergone Notch signaling.

## 6. Transforming Growth Factor Beta (TGF-beta) Family

The transforming growth factor-β family include two distinct signaling pathways known as the TGF-β and BMP (Bone Morphogenetic Proteins) signaling pathways. Either pathway makes use of different SMAD proteins, but have SMAD4 in common to exert transcription of target genes. A third pathway, Activin, shares most of the intracellular regulatory and transcriptional proteins of TGF-β signaling but will not be further discussed in this review. 

TGF-β family signaling is initiated by the binding of either TGF-β, Activin or BMP proteins to their corresponding receptor (Figure 3). To date, there is no consensus regarding ligand-receptor specificity, which might be due to their significant amino acid conservation. In general, an active TFG-β family ligand dimer binding with type II family receptor initiates signal transduction only upon interaction with type I family receptors, also known as Activin receptor-like kinases (ALKs). Consequent to ligand binding, a heteromeric receptor complex is formed to phosphorylate pathway specific cytoplasmic Receptor-regulated Smads (R-SMADs). TGF-β receptor signaling is modulated by SMAD2 and SMAD3, whereas BMP receptor signaling is modulated by SMAD1, SMAD5, and SMAD8 (also known as SMAD9). Subsequently, these R-SMADs form protein complexes with co-SMAD4 which translocate to the nucleus to influence target gene expression together with transcriptional coregulators at the transcriptional activator CREB-binding protein (CBP). Oppositely, Inhibitory Smads (I-SMADs), SMAD6 and SMAD7 regulate the signaling cascade by competing with the R-SMADs for SMAD4 or by ubiquitin recruitment for receptor degradation.

The role of TGF-β signaling in HSC biology is highly debated due to inconsistencies between studies [45]. To date, all TGF-β reporter models originate from the *CAGA* sequence elements first detected in the human *PAI-1* promoter [46] (Table 1). These *CAGA* boxes, bind with the TGF-β /activin pathway-restricted SMAD3/SMAD4 complex but not with the SMAD2/SMAD4 complex [47], resulting in potentially incomplete TGF-β receptor signaling coverage. Additionally, special care should be taken when choosing *CAGA* box reporters for the study of HSC regulation, due to their differing genetic background or lack of reporter activity verification in non-hematopoietic organs. The CAGA_12_-eGFP reporter, for example, is made on a fibrillin-1-deficient BL6/SJL background [29], which has important damaging effects on the extracellular matrices in the body. Accordingly, fibrillin-1 is implied to play an important role in HSC expansion [48], causing the CAGA_12_-eGFP reporter to be unfit for HSC studies. On the other hand, the SBE-lucRT mice are composed out of a trifusion detection protein containing Luciferase, red fluorescent protein (RFP), and thymidine kinase for noninvasive in vivo imaging [31]. The combination of this deep-tissue bioluminescence tracking and single cell reporter imaging could be interesting to study HSC migration during transplantation as well. However, Jackson Laboratory reported the same mouse model to be of C57BL/6J; C57BL/6N mixed genetic background instead of the initially reported C57BL/6J-Tyr^c-2J^ genetic background, which could be a point to consider before using this reporter. 

BMP signaling is indispensable for embryogenesis and the development of precursor tissues related to hematopoiesis [49], making the study of this signaling pathway very difficult due to lethality of the transgenic knock-out models [50]. 

Several BMP reporter models have been published (Table 1), all based on the direct target gene of BMP signaling *Id1* promoter [51]. These reporters are predominantly SMAD1/4 and SMAD5/7 responsive, but to a lesser extend sensitive to SMAD8/4 [32]. Respectively for HSC studies, this sensitivity concern is negligible, for SMAD8 is not expressed in adult murine HSCs [52]. 

As with most of the conserved signaling pathways, the BMP reporters have been mainly used in embryonic and organ development studies. The BRE-gfp reporter is the only BMP reporter used to study adult HSC biology [53]. Two HSC types (BMP activated and non-BMP activated) were detected with equal capacity to assume normal hematopoietic function. BMP activated HSCs originated from the aorta-gonad-mesonephros (AGM) region, but as development progressed and thus migration of HSCs to other anatomical tissues, these HSCs became predominantly non-BMP activated. This demonstrates the importance of the hematological niche and the signals it provides to the residing HSCs. Nonetheless, it is debatable whether the BRE-gfp reporter is the best model to study BMP signaling, due to previous concerns about the integration site of the transgene [33]. Differences in reporter expression were reported between the BRE-LacZ and BRE-gfp reporter [33,34] even though the transgene structures were identical. This inconsistency could possibly be due to CMV enhancer silencing in specific tissues, making these models less attractive. The BRE-GAL mouse model could be a more sensitive option owing to its zinc finger Schnurri (Shn) protein which acts as a cofactor to elicit transcriptional response through a SMAD1 and SMAD4 complex. Its binding requires a five-nucleotide spacing between the 7 BRE binding sites and multimerized xenopus id3 BRE gene driver [35]. Even though the authors mention that their BRE-GAL mouse embryonic cells (mES) only respond to a range of intermediate to high concentration of BMP ligands, they also show that BRE-GAL mECs are more sensitive compared to the BRE-lac1 and BRE-lac2 reporter mice mECs which associate with SMAD1 and SMAD5 signaling [32]. This difference in sensitivity is probably due to the increased BRE copy numbers, yet BRE-lac1 and BRE-lac2 mice with identical constructs also showed differences in sensitivity which strongly hints to genetic insertional influence.

Hence, the TGF-β family is a complex and strongly contextual signaling pathway. Even though the TGF-β and BMP reporters have strictly distinct target genes, upstream cytoplasmic effector molecules cause ambiguous interpretation. SMAD5 for example is not always restricted to BMP signaling since it can also bind to Smad Binding Elements (SBE), suggesting an association with TGF-β signaling. Activin signaling may additionally mask exclusive TGF-β signaling and other activated downstream Smad-independent pathways are feasible options upon receptor binding as well [54,55]. A combination of TGF-β family reporter models could however shed new light on the debate whether these signaling pathways are active during HSC development.

## 7. Hedgehog Signaling

The hedgehog (Hh) signaling pathway is well known for its function in organ patterning during embryogenesis by controlling cell proliferation, differentiation and maturation. Despite it being largely inactive in adult tissues, it has been implicated in several types of cancer. Its potential role in HSC maintenance has been studied to a lesser extent and usually hints towards a synergistic action with other conserved signaling pathways mentioned previously in this review. 

Hh cell signaling is regulated by the GLI-Kruppel proteins (Gli1-3) which act as transcriptional repressors or activators. These zinc-finger containing proteins are highly conserved, but differ at the N-terminal and C-terminal domains which determines their functional role. Generally, Gli1 is known to be a constitutive transcriptional activator, Gli2 can be either a transcriptional activator or repressor, and Gli3 is a transcriptional repressor. The transformation of Gli proteins to their A-Gli (activator) of R-Gli (repressor) form is directed by the protein Suppressor of fused (Sufu) which directs proteasomal cleavage activity of the Gli-protein terminal ends. Gli1 on the other hand, is transcribed depending on the balance between A-Gli2 and R-Gli3. The Hh receptor, Patched (Ptch), responds to three secreted ligands: Sonic, Desert, and Indian (Figure 4). When the pathway is inactivated, Patched acts as an inhibitor of the secondary receptor Smoothened (Smo) by cytosolic sequestration and hindrance of its function in Hh signal activation. Consequently, Gli2 and Gli3 are truncated by proteasomal activity to become transcriptional repressors. Interestingly, two key-regulators of the Wnt signaling pathway, GSK-3β and CK1α, regulate this proteasomal cleavage of Gli2 and Gli3 through ser/thr phosphorylation, similar to β-catenin. R-Gli2 and R-Gli3 are then able to translocate to the nucleus to repress transcriptional activity of Hh target genes. Alternatively, when Hh signaling is active upon ligand binding, Patched together with the co-receptors BOC/CDO abrogate the inhibitor function resulting in the assent of Smoothened to fuse to the cell membrane. Patched attracts several proteins which aid in the dissociation of the Gli2 and Gli3 proteins from Sufu to abrogate subsequent proteasomal cleavage into their repressor form. Gli2 is processed into A-Gli2 which translocates into the nucleus as transcriptional activator to initiate *Gli1* and other target gene expression. Hh signaling is believed to induce different gene sets depending on the ligand activation gradient and tissue [56]. Bai et al. proposed *Gli1* to be a reliable Hh transcriptional target gene, although Gli1 is not required for proper Hh signaling [36]. Nevertheless, Gli1 has the ability to substitute impaired Gli2 function [57], suggesting it to be an important failsafe method in the pathway.

Most studies on Hh signaling have been performed in Drosophila and Zebrafish models in embryonic development. The first known Hh murine reporter was the Gli1^lz^ reporter [36] (Table 1), which is a knock-in of the *Gli1* locus with a *lacZ* reporter protein, resulting in a dysfunctional Gli1. The *Gli1* knock-out is the only non-lethal mutant of the Gli family members. This same reporter was used as a Gli1^null^ (homozygote) model to study the role of Gli1 in adult HSC maintenance [37]. The absence of Gli1 led to an increased proportion of long-term HSCs and improved engraftment after transplantation. Consequently, the loss of proliferation led to the impairment of myeloid differentiation and defective stress hematopoiesis. Other vertebrate Hh reporter models are either the murine CNE1, 6, 9, 10 lacZ mice [38] or the conditional Gli1^CreERT^ x *Rosa26*-EGFP [39] which study the role of Gli3 or Gli1, respectively. Studies on Gli2 and Gli3 have been solely performed with cell line reporter assays or in combination with the Gli1^lz^ reporter [58]. The main reason is that Gli1 is mostly dispensable for normal development, resulting in mildly affected phenotypes when mutated, whereas Gli2 and Gli3 models are generally embryonically lethal [36,59,60]. Even though *Ptch* is considered to be a negative regulator and a possible gene target of the Hh pathway, no murine reporter models are known to date using *Ptch*. Given the possible interference of other pathways on the downstream effector Gli1 [61,62] and the fact that Gli1 mediates *Ptch* expression [37], *Ptch* is a less ideal gene for a Hh reporter.

A noteworthy complication for interpretation of Hh signaling is its multifaceted role and context dependency. Depending on the location of the cell within the ligand excreting niche, a different concentration of Hh signaling could be detected which in turn affects the intricate balance between the repressor and activator functions of the Gli proteins. Consequently, the net-result of R-Gli and A-Gli proteins can drive or prime a distinct set of genes in a tissue specific manner [56]. Especially in the context of HSC differentiation, spatial cell division is thought to be driven by micro-niche cues instructing which daughter cells remain stem cells or progress towards progenitor cells [2]. Altogether, the development of a multifluorescent reporter which visualizes the contribution of all three Gli proteins would be ideal however complicated, especially considering the proteasomal-processing of the Gli proteins.

## 8. Improved Reporter Models

In general, the most feasible reporter systems to date are directed by fluorescent proteins. In the context of signaling pathways, murine reporter models with multi-color hue detection have not been attempted as in cell tracing models. For instance, the lentiviral LeGO vectors have shown to exert multiple fluorescent protein combinations for clonal cell tracking in HSCs [63]. Recently, a murine model “Hue” (comparable to the Brainbow2.1 Confetti mice), studied the clonal HSC distribution with > 103 possible colors [64,65]. Due to the combination of several integrated XFP copies; new colors can be defined by confocal fluorescence intensity profile detection of each XFP independently and of XFP combinations [66,67]. However, due to the importance of stem cell phenotyping and cell sorting in HSC studies, the use of flow cytometry techniques is indispensable and should be carefully considered when designing reporter models. Therefore, well thought strategies could give valuable information on the complex signaling cross-talk by the use of multi-reporter models. To date, solely the double Notch-Wnt reporter (TNR x TOPGAL) has been used in HSC cell signaling studies [24], possibly due to the complicated genetic crossing and breeding strategies it entails to make a multi-reporter. CRISPR-Cas9 transgenesis is thought to make a breakthrough in rapid and effective generation of transgenic models, not to mention it would settle the dilemma of randomly inserted DNA binding sites. Recently, an efficient method to generate gene-targeted mouse models was proposed through tetraploid complementation in EPS cells in just 2 months’ time [68]. 

Before developing suitable murine reporter models, a knowledgeable reporter gene and fluorescent protein strategy must be defined. Caution should be taken when merely interpreting gene expression profiles for the selection of pathway-specific genes; hence they do not reflect actual protein effector function, especially when analyzing a complete signaling pathway. For instance, Gupta showed that *TCF1* expression in the Wnt signaling pathway not always coincided with nuclear TCF1 protein and accordingly TOPGAL expression. Solely in the presence of stabilized β-catenin would *TCF1* expression have a positive reporter expression and thus representation of active Wnt signaling [10]. Therefore, a preference should be employed for knock-in models of pathway specific reporter genes which have been reported to play a role in adult HSC biology, such as *Axin2* (Wnt signaling), *Hes1/5* (Notch signaling), *BRE*, and *CAGA* endogenous promoters (BMP and TGF-β signaling) and *Gli* (Hh signaling). Secondly, the chosen fluorescent protein should match the contextual pathway dynamics while considering the maturation and degradation time (how quickly is it turned on and off). Cai, for instance studied a set of 15 XFPs on in vivo aggregate formation, photostability, and paraformaldehyde fixation stability before opting for a new Brainbow multi-fluorescent toolbox design [69]. Thirdly, the purpose of the reporter model should accompany the intrinsic fluorescent properties of the tissue or cells of interest. Ideally, bone marrow studies are carried out with far-red fluorescent proteins due to their reduced autofluorescence [70], since the bone marrow is a typical organ with high background fluorescence [71,72]. In general, the fluorescent protein and distribution should exceed the background fluorescence of the cells or tissue and have a manageable fluorescent intensity [73].

We envision a fluorescent reporter protein strategy adequate for flow cytometry technology based on the contextual setting of the pathway, brightness and maturation of the XFP [74] and FRET-based biosensory in the case of intrapathway protein interaction (Table 2). Wnt and Notch signaling pathways are thought to cooperate together in HSC maintenance and have been studied with the TNR x TOPGAL reporter [24]. Based on the resulting conclusions, a manageable reporter combination is recommended with compatible fluorescent proteins. The combination of the existing Axin2-mTurquoise model [18] and a proposed Hes1/5-mVenus model would make the simultaneous study of both pathways feasible. mTurquoise2 has been proposed to be a suitable alternative to the β-galactosidase variant of the Axin2-LacZ Wnt reporter due to its high brightness (quantum yield (QY) = 0,93) and easy handling in the ultra-violet laser line (405/407 nm). The co-expression of mTurquoise2 [75] and mVenus [76] have minimal bleed through since their excitation maxima are mutually exclusive (434 nm and 515 nm, respectively), while still providing enough possibilities for additional fluorescent phenotypic markers. mVenus is an interesting yellow-reporter, compared with its counterparts due to its rapid maturation time. However, mVenus is also known to have a half-life ranging around 24 h, which could hint towards a destabilized form of mVenus as a better option for rapid degradation [44]. Especially in the contextual setting of Notch signaling, where dosage effects and oscillations in target gene expression are thought to be important in HSC maintenance and development, a rapid turn-over would be ideal for the study of its quiescence or proliferative action. Furthermore, these fluorescent proteins are well-matching FRET partners, with mTurquoise2 as the donor and mVenus as the acceptor [77]. For a TGF-β and BMP reporter, an *Id1*-mKOκ and *PAI-1*-mScarlet-I transgenic approach would have attractive properties. Considering that BMP signaling is highly present in bone tissue and close to matrix components, a far-red XFP could diminish the existence of autofluorescence for easier investigation of HSCs. Both the orange-protein mKOκ [78] and far-red mScarlet-I [79] are excitable by the yellow-green laser (561 nm), but are easily separated by narrow selected collection windows (i.e., longpass and bandpass filters). mKOκ and mScarlet-I are the only orange/far red-like XFPs with a relatively high brightness (QY = 0.61 and QY = 0.70, respectively). Although mKOκ has a longer maturation time compared to all other proposed fluorescent proteins, its applicability is still within a reasonable range and could be shortened with a PEST domain. The combination of a TGF-β/BMP reporter could shed light on the dual-responsive SMAD5 protein, which up to date has a confusing role in hematopoiesis [80,81,82]. Lastly, the Hh reporter could be well represented by a *Gli1*-mNeonGreen variant. mNeonGreen [83] is a brighter yellow-green protein alternative to GFP and is thought to not recognize GFP antibodies. In combination with other reporter models with GFP-like XFPs, a distinction in antibody recognizing epitopes is very beneficial when fluorescent expression is low. In the case of Hh signaling, where its role in HSC maintenance is believed to be mainly active during stress-hematopoiesis, low reporter expression profiles could be expected during physiological hematopoiesis which may have been missed in the Gli1^lz^ reporter. Even insight into the cross-pathway interaction of Wnt and Hg signaling can be studied with the proposed FRET pairs mTurquoise2 and mNeonGreen [77]. 

The proposed set of reporter mice do not only have individual practicality, but furthermore have been envisaged for the combination of a multi-XFP mouse combining all five pathway reporters. In view of excitation and emission spectra, an accomplishable combination is feasible without unsolvable discriminative measurement of the XFP expression. By careful election of longpass and bandpass filters, all five XFP expression profiles can be evaluated while still being able to use phenotypic makers to determine specific hematopoietic cell subsets. A standard flow cytometer has an ultra-violet (405 or 407 nm), blue (488 nm), and yellow-green (561 nm) laser and the possibility to change filter sets according to the XFP combination. For proper XFP reporter distinction, mTurquoise2 is best excited by 405 or 407 nm, mNeonGreen and mVenus by 488 nm, and mKOκ and mScarlet-I by 561 nm (Figure 5). Possible emission filter sets are: mTurquoise2 (525/50 nm), mNeonGreen (510/10 nm), mVenus (542/27 nm), mKOκ (560/20 nm) and mScarlet-I (610/20 nm). A number of modifications are possible regarding the filter sets, however proper distinction of mNeonGreen and mVenus is best achieved by the proposed bandpass filters [84]. Of course, spectral flow cytometry would make detection even less complex, allowing measurements of several reporter proteins in combination with a multitude of cell surface markers in complex mixtures of cells.

## 9. Conclusions

The regulation of HSC homeostasis or even stress hematopoiesis makes the interpretation of the entirety of involved signaling pathways complicated. For therapeutic purposes, an adequate understanding on how to conduct clinically prepared HSCs to the desired state will only be feasible with more coherent and interconnective understanding on the role of signaling pathways. Merely stimulating one or two signaling routes, as often is done in in vitro HSC priming prior to transplantation, will not solve the equation of the intracellular pathway crosstalk which is strictly regulated in the physiological bone marrow niche. Although several attempts haven been completed to mimic the HSC niche in for example (3D) scaffolds or in bioreactors, researchers have not reached the point of re-enacting the cues required in HSC regulation. Before coming to that accomplishment, more knowledge should be gained on how HSCs respond to HSC niche ligands under physiological and transplantation settings in vivo [87]. In particular for human HSC lympho-myeloid engraftment, humanized immune-deficient xenograft models could provide insight into the discrepancies between human and murine signaling pathway function to clarify translation into clinical application [88]. By the use of well thought out transgenic reporter models in combination with potent XFP technology, much of the lacking information and inconsistencies between existing models can be solved. Intra-pathway interaction is regarded to be of such magnitude and complexity that quantum computing might be a necessary instrument to truly and intuitively comprehend the intricacies of desired HSC priming. In the course of active interpretation of HSC cell signaling, new strategies can be invented for tissue regeneration purposes with a patient’s autologous cells.

## Figures and Tables

**Figure 1 cells-09-02264-f001:**
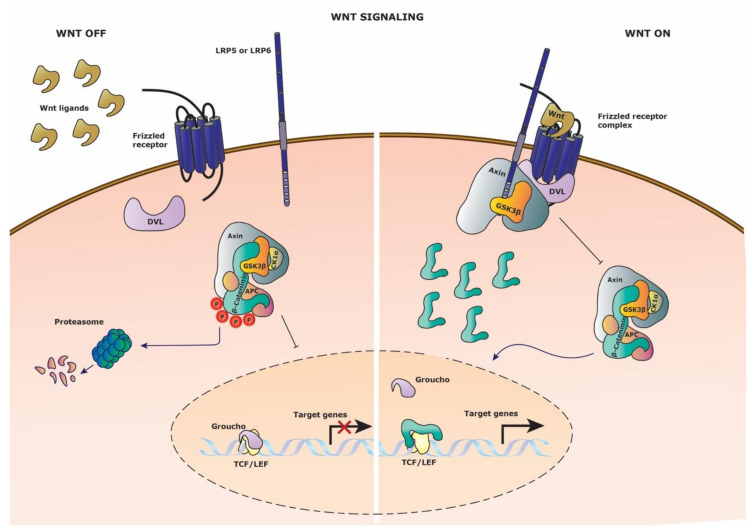
Wnt signaling pathway. Schematic representation of the Wnt signaling pathway. (Left side) Wnt signaling is inactive in the absence of Wnt ligand on the Frizzled receptor. The destruction complex thus can be formed and phosphorylated the constitutively cytosolic β-catenin, leading to proteasomal degradation. Groucho retains its repressor activity by binding the TCF/Lef transcription factors and thus target gene expression is restricted. (Right side) Wnt signaling is activated upon binding of Wnt ligand to the Frizzled receptor and recruitment of the co-receptor LPR5/6. Axin, GSK3-β and DVL (Dishevelled) are recruited to the membrane receptor complex, disrupting the destruction complex. Cytosolic β-catenin translocates to the nucleus to compete with Groucho for TCF/Lef transcription factor binding, leading to target gene expression.

**Figure 2 cells-09-02264-f002:**
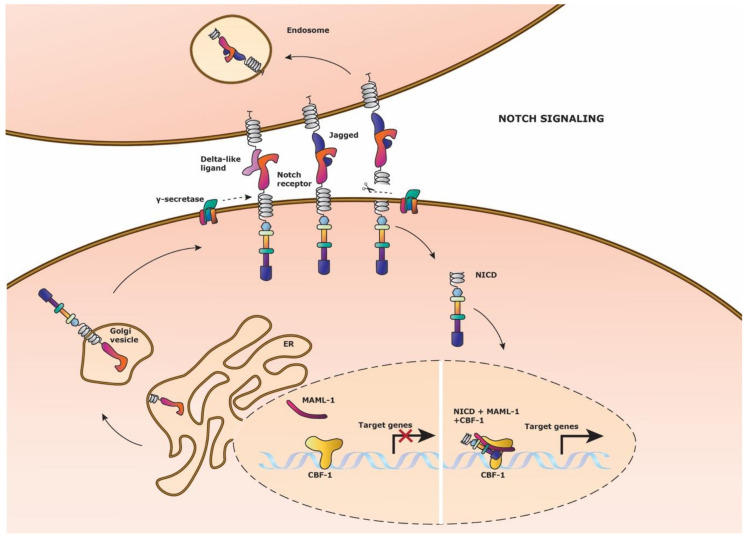
Notch signaling pathway. Schematic representation of the Notch signaling pathway. Notch signaling is activated by Delta-like ligand or Jagged ligand expression on a neighboring cell which binds to the Notch receptor. γ-secretase cleaves off the intracellular domain of the Notch receptor (NICD), which then translocates to the nucleus to form a transcriptional activation complex. NICD and co-activator Mastermind (MAML-1) binds to the DNA binding protein Centromere Binding Factor1 (CBF-1) to initiate Notch target gene expression. (Left side nucleus) After Notch signaling has occurred, the NICD and MAML-1 detaches from CBF-1, to which NICD is recycled into the cytosol. The Notch receptor is replenished by fusion of the newly formed extracellular Notch receptor domain and the NICD on the cell membrane for ligand binding.

**Figure 3 cells-09-02264-f003:**
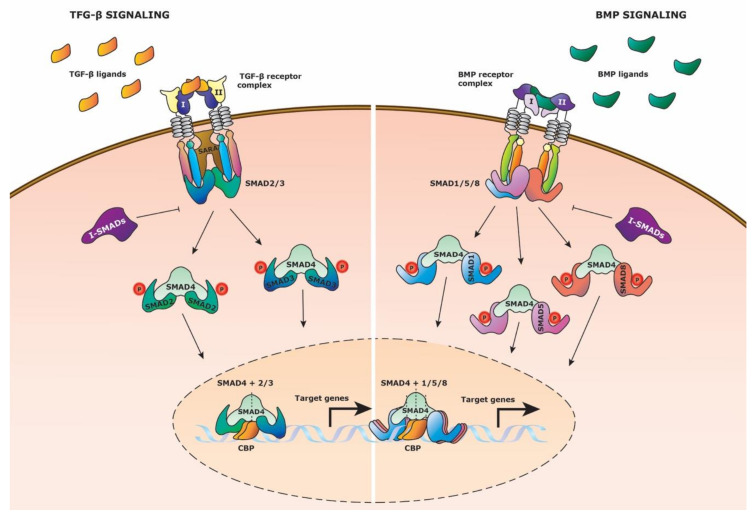
TGF-β family signaling pathway. Schematic representation of the TGF-β family signaling pathway, displaying the TGF-β and BMP pathway. (Left side) TGF-β signaling is activated by TGF-β ligand binding to the TGF-β receptor complex composed out of type I and type II family receptors. SMAD2 and SMAD3 (R-SMADs) bind to the intracellular domain of the receptor complex with the aid of Smad Anchor for Receptor Activation (SARA). Type I receptor then phosphorylates the R-SMADs which causes them to dissociate from the receptor complex. The R-SMADs form separately a complex with co-SMAD4 and translocate to the nucleus to bind to the transcriptional activator CREB-binding protein (CBP) to initiate target gene expression. Inhibitory Smads (I-SMADs) can repress the R-SMADs from binding to co-SMAD4. (Right side) BMP signaling is activated by BMP ligand binding to the BMP receptor complex composed out of type I and type II family receptors. SMAD1, SMAD5, and SMAD8 (R-SMADs) bind to the intracellular domain of the receptor complex. Type I receptor phosphorylates the R-SMADs which causes them to dissociate from the receptor complex. The R-SMADs form separately a complex with co-SMAD4 and translocate to the nucleus to bind to the transcriptional activator CREB-binding protein (CBP) to initiate target gene expression. Inhibitory Smads (I-SMADs) can repress the R-SMADs from binding to co-SMAD4.

**Figure 4 cells-09-02264-f004:**
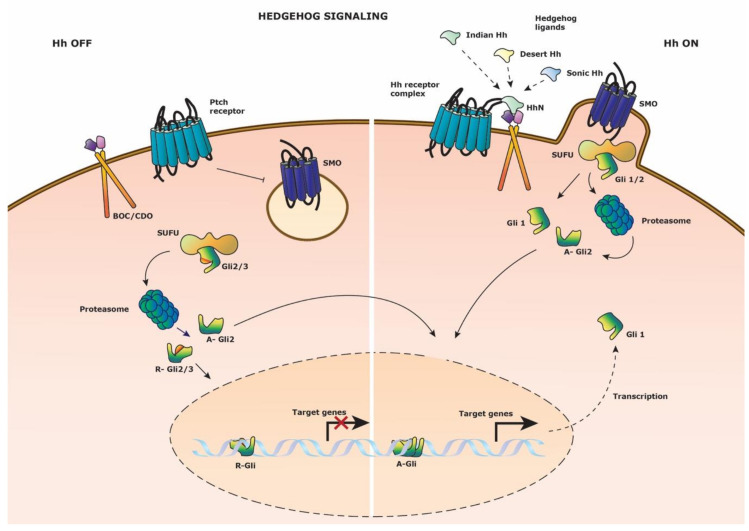
Hedgehog signaling pathway. Schematic representation of the Hedgehog signaling pathway. (Left side) Hedgehog signaling is inactive in the absence of Hedgehog ligand on the Patched (Ptch) receptor. Smoothened (SMO) is restrained from cell membrane anchoring. Suppressor of fused (SUFU) direct the proteasomal cleavage of transcription factors Gli1 and Gli3 to their repressor form; R-Gli2 and R-Gli3. These transcriptional repressors translocate to the nucleus to block target gene expression. (Right side) Hedgehog signaling is activated upon binding of either three of the N-terminal domains (HhN) of the Hedgehog ligands to the Hedgehog (Hh) receptor complex composed out of Ptch and co-receptor BOC/CDO. SMO fuses with the cell membrane and binds SUFU with the bound Gli2 transcription factor. Gli2 is released and undergoes proteasomal degradation to it activator A-Gli2 form which translocates to the nucleus to drive target gene expression. Gli1 is transcribed and reinforces the Hedgehog signaling as a transcriptional activator.

**Figure 5 cells-09-02264-f005:**
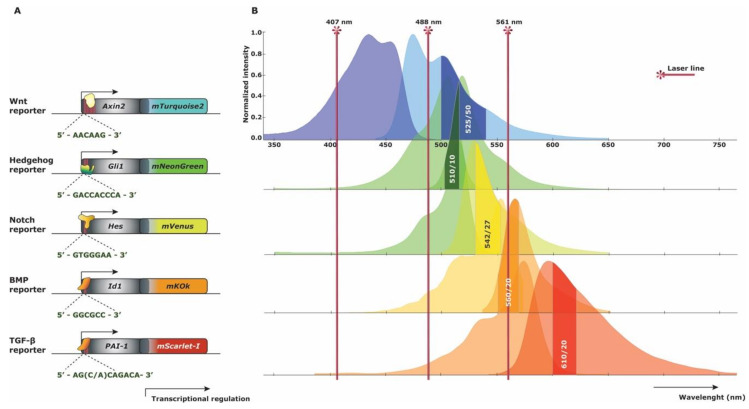
Multi-reporter fluorescent spectra strategy. (**A**) Schematic gene construct representation of new knock-in reporter strategies. The DNA binding motives for the transcriptional activators are represented below each construct. (**B**) Excitation and emission (left and right) spectra of the proposed fluorescent proteins concurrent with the gene constructs for each reporter strategy. Three basic Fluorescent Activated Cell Sorting (FACS) laser lines represent where each fluorescent protein is best excited. Commercially available bandpass filters are depicted in the emission spectra curves of each fluorescent protein, allowing synchronal measurement.

**Table 1 cells-09-02264-t001:** Murine reporters.

Signaling Pathway	Reporter Name	Promoter	Gene Construct	Reporter Protein	Hematopoietic Studies	Reference
Wnt	TOPGAL	minimal *c-fos*	3x TCF/Lef binding sites	*LacZ* (β-galactosidase)	Yes, HSCs	[10]
	Conductin^+/LacZ^	*Axin2* endogenous	Axin2 (8x TCF/Lef binding sites)	nuclear *LacZ* (β-galactosidase)	Yes, adult HSC biology	[8]
	Ax2/d2EGFP	*Axin2* endogenous	Axin2 (8x TCF/Lef binding sites)	d2EGFP	No	[16]
	BAT-gal	minimal-TATA box *siamois*	7x TCF/Lef binding sites	*LacZ* (β-galactosidase)	No	[11]
	TCF/Lef-LacZ	minimal *hsp68*	6x TCF/Lef binding sites	*LacZ* (β-galactosidase)	No	[12]
	ins-TOPEGFP and ins-TOPGAL	minimal thymidine kinase (TK)	6x TCF/Lef binding sites-β-globin HS4 insulators	enhanced GFP or nuclear *LacZ* (β-galactosidase)	Yes, splenic mature T cells during inflammation state	[15]
	LEF-EGFP	minimal *c-fos*	7 Lef-1 binding sites	enhanced GFP	No	[22]
	TCF/Lef:H2B-GFP	minimal *hsp68*	6x TCF/Lef binding sites	H2B-GFP	Partial; embryonic primitive erythroid cells; postnatal thymic medulla	[13]
	Axin2-mTurquoise2	*Axin2* endogenous	Axin2 (8x TCF/Lef binding sites)	mTurquoise2	No	[18]
	Axin2^P2A-rtTA3-T2A-3xNLS-SGFP2^	*Axin2* endogenous	Axin2 (8x TCF/Lef binding sites)	nuclear SGFP2	No	[20]
	TOPeGFP	Minimal thymidine kinase (TK)	6x TCF/Lef binding sites	Enhanced GFP	No	[23]
Notch	TNR (Transgenic Notch reporter)	basal SV40 (simian virus)	4x CBF binding sites	enhanced GFP	Yes, adult HSC biology	[24]
	NAS (Notch Activity Sensor)	minimal TPI (Epstein Barr virus)	12x CBF binding sites	nuclear *LacZ* (β-galactosidase)	No, absent signaling in lymphoid tissues	[25]
	CBF:H2B-Venus	minimal SV40 (simian virus)	4x CBF binding sites	H2B-Venus	No	[26]
	Hes1 and Hes5-GFP	*Hes1* or *Hes5* endogenous	2x CBF binding sites	destabilized enhanced GFP	No	[27]
	Hes1-EmGFPSAT	*Hes1* endogenous	2x CBF binding sites	emerald GFP	Yes	[28]
TGF-β	CAGA12-eGFP	adenovirus major late promoter (MLP)	12x CAGA repeats (PAI-1 promoter)	enhanced GFP	No	[29]
	SBE-luc	thymidine kinase (TK)	12x CAGA repeats (PAI-1 promoter)	Luciferase	No	[30]
	SBE-lucRT	thymidine kinase (TK)	12x CAGA repeats (PAI-1 promoter)	Luciferase, RFP and thymidine kinase (trifusion protein)	No	[31]
BMP	BRE-lac1, BRE-lac2 and BRE-luc	minimal MLP (adenoviral major late promoter)	2x BRE binding sites (Id1 promoter)	*LacZ* (β-galactosidase) or Luciferase	No	[32]
	BRE:gfp	CMV (cytomegalovirus)	2x BRE binding sites (Id1 promoter)	enhanced GFP	Yes, adult HSC biology	[33]
	BRE-LacZ	minimal *Hspa1a*	2x BRE binding sites (Id1 promoter)	*LacZ* (β-galactosidase)	No	[34]
	BRE-GAL	*Id3* (Xenopus)	7x BRE binding sites (Id3 promoter)	nuclear *LacZ* (β-galactosidase)	No	[35]
Hedgehog	Gli1^Lz^	*Gli1* endogenous	*Gli1* (Zinc finger)	*LacZ* (β-galactosidase)	Yes, adult HSC biology	[36,37]
	CNE1, 6, 9, 10	Human *β**-globin*	*Gli3* intronic CNEs	*LacZ* (β-galactosidase)	No	[38]
	Gli1^CreERT^ x *Rosa26*-EGFP	*Gli1* endogenous	*Gli1* (Zinc finger)	enhanced GFP	No	[39]

In vivo murine reporters of the conserved signaling pathways Wnt, Notch, TFG-β, BMP and Hedgehog. The promoter region is either exogenous or endogenous of origin. The gene construct shows the signaling pathway responsive element which are either artificially inserted multimerized binding sites or the endogenous binding sites of a gene. The reporter protein is the detection protein which reports for the active signaling. The hematopoietic studies column comments whether the reporter model has been studied in HSC biology or any hematopoietic subset. Wnt, wingless-related integration site; BMP, bone morphogenic protein; TGF-β, transforming growth factor beta; Lef, lymphoid enhancing factor; TCF, T-cell factor; TOP, TCF/Lef optimal promoter; d2EGFP, destabilized 2 enhanced green fluorescent protein; hsp68, het shock protein 68; GFP, green fluorescent protein; H2B-GFP, histone H2B-green fluorescent protein; mTurquoise2, monomeric turquoise 2; SGFP2, strongly enhanced green fluorescent protein 2; CBF, Centromere-binding protein; TPI, triosephosphate isomerase; SV40, simian virus 40; MLP, Adeno virus mayor late promoter; BRE, B recognition element; Id1, Inhibitor of DNA binding 1; Hspa1a, heat shock protein 1a; Id3, Inhibitor of DNA binding 3; TF, thymidine kinase; Gli1, GLI family zinc finger 1; CNE, conserved non-coding sequences; EGFP, enhanced green fluorescent protein.

**Table 2 cells-09-02264-t002:** Multi-reporter strategy.

Signaling Pathway	Fluorescent Protein	Oligomerization	Quantum Yield	Maturation Time (minutes)	FRET Pair	Reference
Wnt	mTurquoise2	Monomer	0.93	33.5	mTurquoise2-mVenus	[75,85]
Notch	mVenus	Monomer	0.64	17.6	mVenus-mKOκ	[76,86]
BMP	mKOκ	Monomer	0.61	108	mKOκ-mTurquoise2	[77,78]
TFG-β	mScarlet-I	Monomer	0.7	36	mTurquoise2-mScarlet-I	[77,79]
Hedgehog	mNeonGreen	Monomer	0.8	10	mNeonGreen-mTurquoise2	[77,83]

Proposed multi-reporter strategy of the of the conserved signaling pathways Wnt, Notch, BMP, TGF-β and Hedgehog. Each proposed fluorescent protein is based on monomeric oligomerization to prevent protein agglomeration and has a high brightness (quantum yield) and acceptable protein maturation time after transcription. Additionally, the same fluorescent protein combinations can be used for FRET experiments to study signaling pathway protein interaction. Wnt, wingless-related integration site; BMP, bone morphogenic protein; TGF-β, transforming growth factor beta; mTurquoise2, monomeric turquoise 2; mVenus, monomeric Venus; mKOκ, monomeric Koκ, mKusabira-Orange-kappa; mScarlet-I, monomeric scarlet 1; mNeonGreen, monomeric neon green.

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
