# Peer review of "Cell Signaling Pathway Reporters in Adult Hematopoietic Stem Cells"

_cells, 2020, doi:10.3390/cells9102264_

Round 1

Reviewer 1 Report

 The review article entitledCell Signaling Pathway Reporters In Adult Hematopoietic, by de Roo et al. is a comprehensive and easy to follow look at key past achievements and long-term goals of the field in regard to HSC lineage tracers. Overall, the paper covers most of the aspects currently developed. It is written by an internationally recognized research group that has made recent seminal contributions to the field.

The organization of the subsections is logical, and the review covers a fair splay of important studies at a reasonable depth. In most instances, care is also taken to spell out pitfalls and limitations without diminishing achievements or promoting one line of research as superior.

Altogether, this is a very timely summary of progress in the field and ongoing areas of optimization that will be a useful resource for the hematology community as well as the unfamiliar reader.

Author Response

We thank the reviewer for the positive remarks. Based on the comments made by reviewer 1 and 2, we have shortened sections of the manuscript, have more critically reviewed certain aspects of reporter systems and adapted the figures as well as corrected spelling errors

Reviewer 2 Report

This manuscript is a very well written piece of review and offers a lot of important information on the use of different reporters to assay certain developmental signalling pathways in haematopoiesis. In addition, the authors did not just summarize what is known on this subject but importantly they gave their critical points of view. It would benefit immensely for the readers if the authors can address these two main points to improve this nice piece of work before publication:

1-since not all the reporters were used to study haematopoiesis/HSC biology and not all of them were mentioned in the text too, this reviewer suggests to add another column in Table I, with “Notes” or “Additional Information” where the following information should be added: a) used or not used to study haematopoiesis/HSC biology; b) advantages/disadvantages of the reporter.

2-although it reads very well, but this manuscript is far too long, over extensive: sections 8, 9, 10 and 11 are far too generic topics and most of them can be eliminated. The authors can simply take some of the pertinent aspects from these sections (some of the examples given) and integrate them into other parts where is appropriate, in particular from section 11. Sections 8-10 can all be eliminated.

Author Response

We thank the reviewer for the positive remarks. Based on the comments made by reviewer 1 and 2, we have shortened sections of the manuscript, have more critically reviewed certain aspects of reporter systems and adapted the figures as well as corrected spelling errors. Specifically in response to reviewer 2, we have adapted table1 to include whether the reporter was used in blood cells. We have refrained from a strong disadvantage or advantage statement as many reports have not been tested side by side and while one report may not work well for a detain application it may be just right for another. We have made a remark that for most applications in vivo reporters with multimerized binding sites of transcription factors are less desirable.

We have adapted the sections 8, 9, 10, and 11 according to the reviewer’s comments and mostly deleted them.